

# The microbiome profiling of fungivorous black tinder fungus beetle *Bolitophagus reticulatus* reveals the insight into bacterial communities associated with larvae and adults

Agnieszka Kaczmarczyk-Ziemba[1], Grzegorz K. Wagner[2], Krzysztof Grzywnowicz[3], Marek Kucharczyk[4] and Sylwia Zielińska[5,6]

[1] Department of Genetics and Biosystematics, Faculty of Biology, University of Gdansk, Gdansk, Poland
[2] Department of Zoology, Maria Curie-Sklodowska University, Lublin, Poland
[3] Department of Biochemistry, Maria Curie-Sklodowska University, Lublin, Poland
[4] Department of Nature Protection, Maria Curie-Sklodowska University, Lublin, Poland
[5] Department of Bacterial Molecular Genetics, Faculty of Biology, University of Gdansk, Gdansk, Poland
[6] Phage Consultants, Gdansk, Poland

Corresponding author
Agnieszka Kaczmarczyk-Ziemba,
agnieszka.kaczmarczyk@biol.ug.edu.pl

## ABSTRACT

Saproxylic beetles play a crucial role in key processes occurring in forest ecosystems, and together with fungi contribute to the decomposition and mineralization of wood. Among this group are mycetophilic beetles which associate with wood-decaying fungi and use the fruiting body for nourishment and development. Therefore, their feeding strategy (especially in the case of fungivorous species) requires special digestive capabilities to take advantage of the nutritional value of fungal tissue. Although polypore-beetle associations have been investigated in numerous studies, detailed studies focusing on the microbiome associated with species feeding on fruiting bodies of polypores remain limited. Here we investigated the bacterial communities associated with larvae and adults of *Bolitophagus reticulatus* collected from *Fomes fomentarius* growing on two different host tree: beech (*Fagus* sp.) and birch (*Betula* sp.), respectively. Among 24 identified bacterial phyla, three were the most relatively abundant (Proteobacteria, Actinobacteria and Bacteroidetes). Moreover, we tried to find unique patterns of bacteria abundances which could be correlated with the long-term field observation showing that the fruiting bodies of *F. fomentarius,* growing on birch are more inhabited by beetles than fruiting bodies of the same fungus species growing on beech. Biochemical analyses showed that the level of protease inhibitors and secondary metabolites in *F. fomentarius* is higher in healthy fruiting bodies than in the inhabited ones. However, tested microbiome samples primarily clustered by developmental stage of *B. reticulatus* and host tree did not appear to impact the taxonomic distribution of the communities. This observation was supported by statistical analyses.

## INTRODUCTION

Saproxylic beetles are directly or indirectly related to wood during at least one developmental stage (*Speight, 1989*). They play a critical role in key processes occurring in forest ecosystems and together with fungi, contribute to the decomposition and mineralization of wood (*Gutowski & Buchholz, 2000*). These species are a major component of forest biodiversity and help to maintain a specific homeostasis of the ecosystem. Saproxylic beetles can occupy several ecological nishes and therefore xylophages, cambiophages, predators, necrophiles, and finally mycetophiles can be distinguished (*Gutowski et al., 2004*).

Mycetophilic beetles associate with wood-decaying fungi and use the fruiting body for nourishment and development (*Gutowski, 2006*). Their feeding strategy requires special digestive capabilities to take advantage of the nutritional value of fungal tissue. Associated microorganisms play a crucial role in those processes. Although polypore-beetle associations have been investigated in numerous studies (*Nikitsky & Schigel, 2004*; *Schigel, Niemelä & Kinnunen, 2006*; *Schigel, 2009*; *Schigel, 2011*; *Schigel, 2012*), the detailed projects focused on their microbiome remain limited. Previous study was focused rather on microbiota associated with fungivorous insects (not only beetles, but also ants and termites) which cultivate fungi for food to take advantage of the nutritional value of fungal tissue (*Aylward et al., 2014*). However, the pilot investigations of microbiome of beetles associated with wood-decaying fungi also have been initiated. Recently, *Wiater et al. (2018)* identified bacteria *Paenibacillus* sp. in the gut of fungivorous darkling beetle *Diaperis boleti* (Tenebrionidae) feeding on polypore fungus *Laetiporus sulphureus*. These bacteria effectively degrade fungal $\alpha$-(1 →3)-glucan present in cell wall of fungi. More complex studies focused on profiling the microbiome of fungivorous beetles have not been performed yet.

The black tinder fungus beetle *Bolitophagus reticulatus* (Tenebrionidae) is a fungivorous species occurring widely throughout European forests (Fig. 1A). This beetle belongs to tribe Bolitophagini which represent the feeding strategy of dwellers. Larvae of beetles described as dwellers are fungivorous. In turn, their adults spend most of their life cycle inside the fruiting body and leave the fungus usually for mating and dispersal only (*Schigel, Niemelä & Kinnunen, 2006*). *B. reticulatus* lives in close association with the perennial basidiocarps of *Fomes fomentarius* (L.) Fr. (commonly known as *the tinder fungus*; Fig. 1B) at all developmental stages and seems to be monophagous on this fungus species (*Midtgaard, Rukke & Sverdrup-Thygeson, 1998* and references therein).

The long-term field observations have shown that *B. reticulatus* is more often found inside *F. fomentarius* fruiting bodies growing on birch (*Betula* sp.) compared with those growing on beech (*Fagus* sp.). Moreover, polypores growing on beech trees are much larger and less inhabited by insects than fruiting bodies growing on birch (*Wagner, 2018*). The growth of fungi is closely correlated with the amount of catechins utilized (*Arunachalam et al., 2003*). Catechins can be taken and metabolized mainly by wood degrading fungi (*Rayner & Boddy, 1988*). Derivatives of catechins are also present in fungi themselves (*Zhou & Liu, 2010*). *Schwarze, Engels & Mattheck (2000)* has shown that the mycelium growing on the

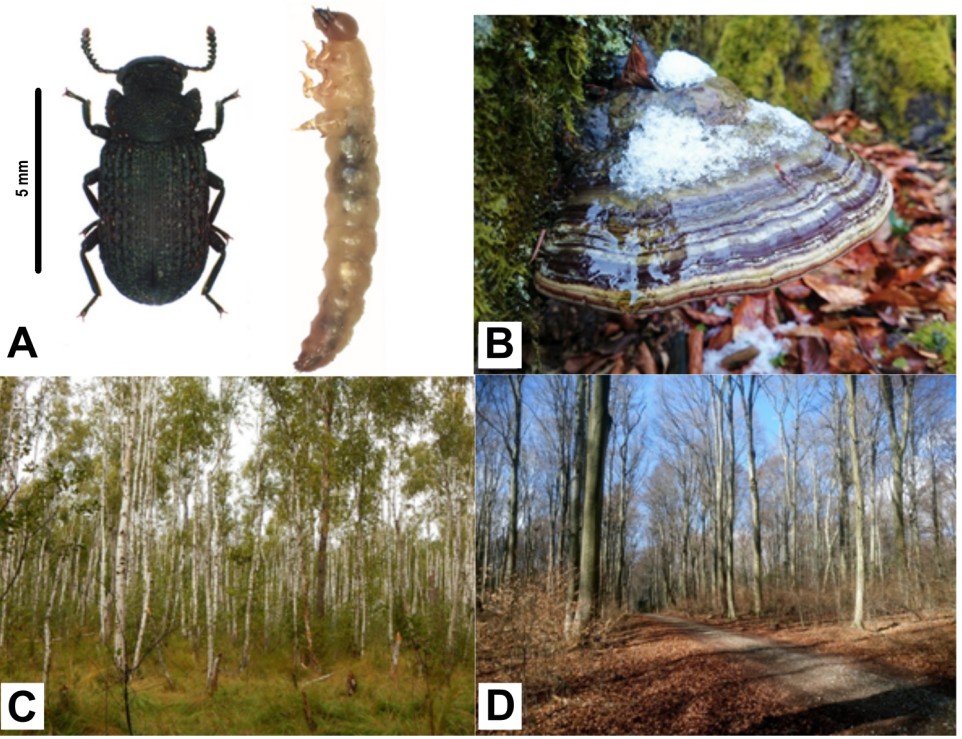

**Figure 1** *Bolitophagus reticulatus* **individuals and** *Fomes fomentarius* **fruiting body with pictures of sampling sites where they were collected.** (A) adult and larva of *Bolitophagus reticulatus* L.; (B) fruiting body of *Fomes fomentarius* (L.) Fr.; (C), swampy birch forest, Poleski National Park; (D), Carpathian beech forest, Roztocze National Park (phot. G. K. Wagner).

tree accumulates the secondary metabolites of its host, especially in the parts covering the fruiting bodies. This process may be correlated with observed differences in colonization degrees. Fungal metabolites are of considerable interest and remarkable importance as new lead compounds for plant and animal or human protection. Importantly, fungal polyketides are one of the largest and most structurally diverse classes of naturally occurring compounds, ranging from simple aromatic metabolites to complex macrocyclic lactones. They are inhibitors of enzymes, including proteases (*Shen et al., 2015*). However, the state of knowledge about the biological activity of substances derived from *F. fomentarius* remain limited. Antimicrobial activity of extracts derived from the tinder fungus has been described by *Dresch et al. (2015)*, while *Chen et al. (2008)* found that its exopolysaccharide (EPS) has direct antiproliferative effect *in vivo*. In turn, some active compounds (e.g., catalase, extracellular laccase, manganese-dependent peroxidase, carboxymethyl cellulose or xylanase) have been identified by *Jaszek et al. (2006)* and *Elisashvili et al. (2009)*. Nevertheless, there is still little known about the secondary metabolites and inhibitors of proteases, especially with regard to fruiting bodies from the natural environment, not from *in vitro* cultures. Therefore, more detailed and complex studies are needed to test the hypothesis of the correlation between profiles of the secondary metabolites and inhibitors of proteases, and the degree of polypores colonization.

In the present study, we investigated the bacterial communities associated with larvae and adults of *B. reticulatus* collected from *F. fomentarius* growing on two different host tree: beech and birch, respectively. We used the Next Generation Sequencing (NGS) of the 16S rRNA gene to define whether the bacterial communities vary among the two tested developmental stages of *B. reticulatus*. Lastly, we investigated the potential differences between microbiome profiles of individuals collected from the tinder fungus growing on birch and beech and combined the results with those obtained from studies on the biochemical composition of the *F. fomentarius* fruiting bodies growing on wood of two different tree species.

## MATERIALS & METHODS

### Study area and sample collection

Materials for the study were collected in two National Parks: Poleski NP and Roztocze NP in south-eastern Poland. Field studies were approved by the Ministry of the Environment in Poland (field study approval numbers: DLP-III-4102-21/1728/15/MD for the field study in Poleski National Park and DPL-LLL-4102-609/1699/14/MD for the field study in Roztocze National Park). Adults and larvae of black tinder fungus beetle *B. reticulatus* were caught in fruiting bodies of *F. fomentarius* (Figs. 1A and 1B, respectively). Five adults and five larvae were collected from the same fruit body growing on fallen birch trunk in Łowiszów (DMS: 51°26′57.762″N, 23°14′29.839″E), 10. November 2016, Poleski NP (Fig. 1C) and the same number of individuals were collected from one fungus growing on fallen beech stump in Obrocz (DMS: 50°34′32.403″N, 23°′24.388″E), 10. December 2016, Roztocze NP (Fig. 1D). Specimens were separately placed into tubes and stored at −30 °C. Afterwards, the tubes with insects were sent for further analyses to the Department of Genetics and Biosystematics, University of Gdansk, Poland.

Studies on the occurrence of *F. fomentarius* fruiting bodies growing on birch and beech were performed during the years 2013–2016. In those studies, four fallen tree trunks from each tree species were selected for further analyses. Chosen trunks were plentifully covered with sporocarps. Beech trunks were located in Roztocze NP (localization—Obrocz, DMS: 50°34′32.403″N, 23°0′24.388″E), while birch trunks were located in Poleski NP (localization: Lipniak DMS: 51°27′51.363″N, 23°6′28.062″E). Healthy and settled fruiting bodies were counted in the field and results are presented in Table S1. For biochemical analyses, fruiting bodies of the tinder fungus (Fig. 1B) were collected in July 2016 from fallen trunks of each tree species growing in two mentioned localities. In both sampling sites, five samples of healthy fruiting bodies and the same number of the inhabited fruiting bodies were taken. The fruiting bodies were inhabited by few species of mycophagous beetles with dominance of the studied species (*B. reticulatus*). Samples were cut out from the tissue above the hymenium.

### DNA extraction

DNA was extracted from the whole bodies of *B. reticulatus* at two developmental stages (larvae and adults, respectively). Insects were rinsed three times in sterile distilled water prior to DNA extraction without soaking in ethanol. Specimens were homogenized

with 0.7 mm garnet beads in a high speed benchtop homogenizer FastPrep®-24 (MP Biomedicals, Santa Ana, CA, USA) at 4.5 m/s for 20 s. The total DNA was then extracted using the Sherlock AX Purification Kit (A&A Biotechnology, Gdynia, Poland). Thus, the presented study resolves the complex microbial population structure of two developmental stages of *B. reticulatus* collected from fruiting bodies growing on different hosts. To avoid cross contamination of samples, the process was performed with sterile equipment. The quantity and quality of the extracted DNA were evaluated by using a Nano Drop ND-1000 spectrophotometer (Nano Drop Technologies, Wilmington, DE, USA). After extraction, the DNA was stored at −20 °C until further use. Twelve samples consisting of genetic material isolated from larvae and adults (one individual per DNA isolate and three isolates per developmental stage collected from fungi growing on different hosts) were used for microbiome analyses.

## 16S rRNA gene sequencing and bacterial community analyses

The V3-V4 hypervariable regions of bacterial 16S rRNA gene region were amplified using the following primer set: 341F-5′-CCTACGGGNGGCWGCAG-3′ and 785R-5′-GACTACHVGGGTATCTAATCC-3′. The targeted gene region has been shown to be suitable for the Illumina (San Diego, CA, USA) sequencing (*Klindworth et al., 2013*). Libraries were prepared with a two-step PCR protocol based on Illumina's ''16S metagenomic library prep guide'' (15044223 Rev. B) with NEBNext® Q5 Hotstart High-Fidelity DNA polymerase (New England BioLabs Inc., Ipswich, MA, USA) according to the manufacturer's protocol, using Q5® Hot Start High-Fidelity 2X Master Mix (NEBNext; New England BioLabs) and the Nextera Index kit (2 × 250 bp). PCR was carried out under the following conditions: 98 °C for 30 s for initial denaturation of the DNA, followed by 25 cycles of 98 °C for 10 s, 55 °C for 30 s, 72 °C for 20 s and for additional 2 min at 72 °C for final extension. Paired-end (PE, 2 × 250 nt) sequencing with a 5% PhiX spike-in was performed with an Illumina MiSeq (MiSeq Reagent kit v2) at Genomed, Warsaw, Poland; following the manufacturer's run protocols (Illumina, Inc., San Diego, CA, USA). Automatic primary analysis and de-multiplexing of the raw reads were performed with MiSeq, with the use of MiSeq Reporter (MSR) v2.6 (16S Metagenomics Protocol).

The genetic material isolated from 12 individuals was sequenced separately. Samples were then marked as follows: L-*Fagus*-X and Im-*Fagus*-X for larva and adult collected from *F. fomentarius* fruiting body growing on beech stump; L-*Betula*-X and Im-*Betula*-X for larva and adult collected from *F. fomentarius* fruiting body growing on birch stump (X indicates the number of individual).

The samples were processed and analyzed using the Quantitative Insights Into Microbial Ecology (QIIME, version 1.9.1) pipeline (*Caporaso et al., 2010*). Paired-end reads from MiSeq sequencing were quality trimmed and joined with PANDAseq version 2.8 (*Masella et al., 2012*) with a quality threshold of 0.9. The sequences that did not meet the quality criteria were removed from further analysis (mean quality >20). Chimeric reads detection was performed with VSEARCH, version 1.7.0 (*Rognes et al., 2016*), an open-source replacement of USEARCH software. Operational Taxonomic Unit (OTU) clustering was performed using UCLUST method, version 1.2.22q (*Edgar, 2010*) and taxonomic

assignment performed at 97% against the SILVA v.132 database (*Quast et al., 2013*). The Biological Observation Matrix (BIOM) table was used as the core data for downstream analyses (*McDonald et al., 2012*). Any sequences that were classified as Mitochondria or Chloroplast, as well as singletons were filtered out of the dataset. Sample reads were rarefied to 38,188 reads. OTU saturation was evaluated with rarefaction curves using Chao1 richness estimate. Moreover, the diversity indices were estimated, including the Chao1, PD (a quantitative measure of phylogenetic diversity), Shannon, and Simpson indices and also the number of observed OTUs. Comparison of the microbial community structures was performed with the use of UniFrac (*Lozupone & Knight, 2005*) and Emperor (*Vázquez-Baeza et al., 2013*). A two-sample $t$-test with a non-parametric Monte Carlo permutations ($n = 999$) and Bonferroni correction was used to test for statistically significant difference in alpha diversity between predefined groups (according to (1) developmental stage, (2) host tree species, (3) both mentioned factors). A two-dimensional Principal Coordinate Analysis (PCoA) was conducted from weighted UniFrac distances obtained in previous steps. In order to determine if observed clusters of samples were significantly dissimilar, an analysis of similarity (ANOSIM; *Clarke, 1993*) was performed in QIIME with 999 permutations.

Similarity percentage (SIMPER) analysis was performed to calculate the average dissimilarities in microbial community structures between particular samples and to access which family was responsible for the observed differences. Statistical analyses were performed using PAST 3.16 software (*Ryan et al., 2001*). Finally, to illustrate the most abundant bacterial families and community relationships across tested samples a heatmap and dendrogram was generated with Bray-Curtis dissimilarity index. Bacterial families whose relative read abundance was less than 3% of at least one sample were removed. Those analyses were performed in R v.3.4.3 (*R Core Team, 2017*; *Neuwirth, 2014*; *Ploner, 2015*; *Oksanen et al., 2018*; *Warnes et al., 2019*).

## Data availability

Bacterial 16S reads for each sample were submitted to the European Nucleotide Archive (ENA) database under accession number PRJEB23388.

## Biochemical analyses

With the use of thin layer chromatography (TLC), comparison of entomotoxic and insecticidal features of fruiting bodies of *F. fomentarius*, enzyme analyses of the level of protease inhibitors (*Sabotič, Ohm & Künzler, 2016*) and analyses of secondary metabolite profiles was performed (*Anke & Sterner, 2002*).

Samples from 20 sporocarps of a known type (from beech and birch, healthy and inhabited by beetles separately) were mechanically ground and then homogenized in distilled water (for inhibitor determinations and TLC analyses of secondary metabolites) or in methanol (to TLC of secondary metabolites) in a Potter homogenizer; 100 mg of shredded the sporocarp in 5 ml of water or methanol. The homogenates were then centrifuged to give supernatants as assay preparations (*Sobczyk, 2010*; *Jaruga, 2013*). Protein in water extracts was determined by standard Bradford method (*Bradford, 1976*).

The level of protease inhibitors was determined according to *Sobczyk (2010)* and with marker proteases (used to determine type of inhibitor and specific pH) according to *Anson*

*(1938)*. 0.1 ml of the preparation was incubated with 0.1 ml marker enzyme solution (pepsin at pH 5.0 for aspartate acid protease inhibitors, trypsin and papain at pH 7.0 for neutral serine and cysteine protease inhibitors and trypsin at pH 9.0 for alkaline serine protease inhibitors) for 30 min at 37 °C. After this time, 0.5 ml of the hemoglobin solution was added in buffer of appropriate pH and incubated for 1 h at 37 °C. The reaction was stopped by adding 2.0 ml of 5% TCA (trichloroacetic acid). Samples were centrifuged and their absorbance measured at 280 nm. As controls, water instead of preparation (sample for % inhibition calculation) and marker enzyme with water instead of specimen (zero-sample to reset the spectrophotometer) were used. Percent inhibition was determined as the percentage of marker enzyme inhibition (*Sobczyk, 2010*).

During qualitative analysis of secondary metabolites by TLC, two types of extracts from the fruiting bodies were analyzed: aqueous and methanol. TLC chromatography was developed in two systems—ethanol: water (7:3) and ethyl acetate: acetic acid: water (2:1:1). Merck ready TLC plates (type: TLC Silica gel 60 F254) were used. Visualization of plates was done by UV light observation (254 nm and 365 nm), showing visible spots of secondary metabolites. Qualitative analysis were made by spot diameter and intensity of glaring, relative rating (standard TLC procedures). Then calculation of their retardation factor (Rf, defined as the ratio of the distance traveled by the center of a spot to the distance traveled by the solvent front) and estimation of their relative UV intensity was performed (*Jaruga, 2013*). To determine the identity of the compounds the Rf values were compared to the Rf value of compounds listed in databases (*Clevenger et al., 2017*).

## RESULTS

### General description of 16S rRNA gene sequencing results

For each *B. reticulatus* sample, we obtained >38,000 good quality 16S rRNA gene sequences (V3-V4 region), ranging between 38,188 for L-*Betula*-2 and 73,856 for Im-*Betula*-3. Rarefaction curves with Chao1 diversity indices, indicating insect microbiome sampling depth and saturation is shown in Fig. 2A. Observed curves almost reached a plateau at this sequencing depth, suggesting that the sequencing was sufficient for microbiome characterization. More details for sequence data for each sample, as well as the number of the observed OTUs and the diversity indices are shown in Table 1. At least 319 OTUs, ranging from 319–1,223, were observed in different samples of *B. reticulatus*, which indicates that the microbial population is complex.

In all samples, at least 98,94% of the reads could be classified to phylum level. Detailed taxonomic analyses on different ranks are available in supplementary data as sunburst charts (Data S1) and also in a table (Table S2).

### Bacterial community composition

The analysis of bacterial community showed that for both larvae and adults of *B. reticulatus* >99.94% of total reads were represented by Bacteria (Data S1 and Table S2). The remaining percentage comprised Archaea. The microbiomes tested in this study contained 24 phyla. The most abundant phyla across all tested stages were Proteobacteria, Actinobacteria and Bacteroidetes. In each sample of *B. reticulatus* developmental stages collected from

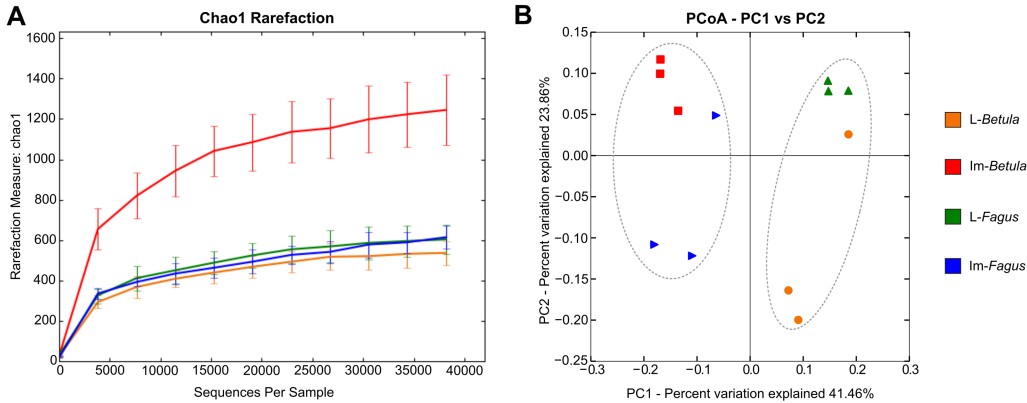

**Figure 2** **Rarefaction analysis and Principal Coordinate Analysis (PCoA) plot of the tested samples.**
(A) Rarefaction curves with Chao1 diversity indices, indicating insect microbiome sampling depth and
saturation. (B) PCoA of bacterial communities associated in tested specimens based on weighted UniFrac
distances. The dotted line indicates the sample clustering according to the developmental stage.

**Table 1** **Summary of the sequencing data and statistical analysis of bacterial communities.**

| ID | No. of bacterial reads | Average length (bp) | No. of observed OTU's | Chao1 index | Shannon index | Simpson index |
|---|---|---|---|---|---|---|
| L-*Betula*-1 | 53,423 | 454 | 463 | 552 | 5.86 | 0.95 |
| L-*Betula*-2 | 38,188 | 451 | 319 | 453 | 5.30 | 0.94 |
| L-*Betula*-3 | 71,060 | 450 | 527 | 594 | 6.04 | 0.96 |
| Im-*Betula*-1 | 71,537 | 452 | 918 | 1017 | 5.96 | 0.95 |
| Im-*Betula*-2 | 59,894 | 452 | 1,195 | 1,395 | 6.96 | 0.98 |
| Im-*Betula*-3 | 73,856 | 452 | 1,223 | 1,236 | 6.72 | 0.97 |
| L-*Fagus*-1 | 70,693 | 450 | 486 | 641 | 6.01 | 0.94 |
| L-*Fagus*-2 | 47,193 | 449 | 433 | 573 | 6.39 | 0.98 |
| L-*Fagus*-3 | 48,758 | 450 | 576 | 725 | 6.16 | 0.96 |
| Im-*Fagus*-1 | 49,161 | 454 | 432 | 524 | 4.90 | 0.90 |
| Im-*Fagus*-2 | 51,603 | 454 | 485 | 554 | 5.20 | 0.92 |
| Im-*Fagus*-3 | 62,315 | 450 | 571 | 624 | 5.59 | 0.94 |
| Total | 697,681 | 451 | 2,527 | 741 | 5.92 | 0.95 |

**Notes.**
The ID abbreviations are defined in text. The number of OTUs (operational taxonomic units) was generated at the 97% se-
quence similarity cut-off. Diversity indices represent the randomly selected subsets for each sample normalized to 38,188 se-
quences.

*F. fomentarius* fruiting bodies growing on both tree species, those phyla jointly accounted for
more than 81.45% of the total microbial sequences obtained. Other bacterial and archaeal
phyla were present in tested microbial communities with different relative abundances
given in Table S2.

The most abundant classes among bacterial communities of all samples were
Alphaproteobacteria, Actinobacteria and Gammaproteobacteria, accounting for more than
69% of the total reads (Data S1 and Table S2). The most abundant orders among analyzed
microbiome profiles were Corynebacteriales, Betaproteobacteriales and Rhizobiales,

accounting for more than 30% of the total reads. The most abundant family was Burkholderiaceae and among that family the most abundant genus was *Burkholderia-Caballeronia-Parabirkholderia*, which accounted for 16.17% of the reads on average (ranging from 0.82% in L-*Fagus*-2 to 38.87% in Im-*Fagus*-1).

Similarities among the bacterial community structures associated with tested samples are illustrated with a heatmap (Fig. 3). We identified 27 families, which relative abundance was not less than 3% of at least one sample. Tested samples primarily clustered by developmental stage of *B. reticulatus* and host tree did not appear to impact the taxonomic distribution of the communities (Fig. 2B). This observation was supported by statistical analyses (ANOSIM: $R = 0.88$, $p = 0.002$, and the alpha diversity indices were only significantly different ($p = 0.004$) between samples grouped according to the developmental stage). In bacterial communities of larvae Microbacteriaceae and Rhiziobiaceae were slightly more abundant, whereas Acidobacteriaceae, Sphingomonadaceae, Rhodanobacteraceae, Mycobacteraceae and Sphingomonadaceae were more abundant in microbiome of *B. reticulatus* adults. SIMPER analysis showed that the last four families were primarily responsible for the differences between microbial communities of larvae and adults.

Analyses of bacterial communities associated with *B. reticulatus* revealed the presence of known endosymbionts. *Wolbachia* has been detected in 4 samples: all larvae collected from *F. fomentarius* growing on beech (0.08% for both L-*Fagus*-1 and L-*Fagus*-2, and 0.14% for L-*Fagus*-3) and in one adult related to birch (<0.01% for Im-*Betula*-2). Moreover, two other known endosymbiotic bacteria have been detected: *Arsenophonus* (0.06% for Im-*Fagus*-3 and L-*Betula*-2) and *Candidatus Cardinium* (0.01% for Im-*Betula*-1).

## Level of protease inhibitors and secondary metabolite profiles in *F. fomentarius* fruiting bodies

The results of biochemical analyses of the collected fruiting bodies showed that the level of inhibition was higher for aspartic acid protease inhibitors in fungus from beech, and for cysteine inhibitors and serine neutral proteases in fungus from birch (Table 2). In the case of inhibitors of basic serine proteases, the levels from both trees were similar (Table 2). The level of inhibitors is higher for healthy fruiting bodies than for the inhabited ones.

Positive TLC results were obtained for both types of extraction (water and methanol) (Table 3), but only in the ethanol-water system. In the ethyl acetate-acetic acid-water system, no separation was obtained and only a spot corresponding to the initial application on the TLC plate was apparent. The strongest spot of secondary metabolites, 0.11176, from a healthy beech tree, has not been identified. There were no differences in the intensities of the weaker 0.9092 spot from the healthy beech tree across samples and the 0.8235 spot, which was present in all fruiting bodies. There is a visible decrease in the intensity of spots 0.7058, 0.1176 and 0.7882 on beech between healthy and colonized fruit bodies. Spots 0.11176 (methanol), 0.9092 (water) and 0.7882 (water) are absent in the fruiting bodies growing on the birch, both healthy and inhabited. The 0.7058 spot in all samples from the aqueous extract did not change its intensity, in comparison to spots with the same Rf from the methanol extract.

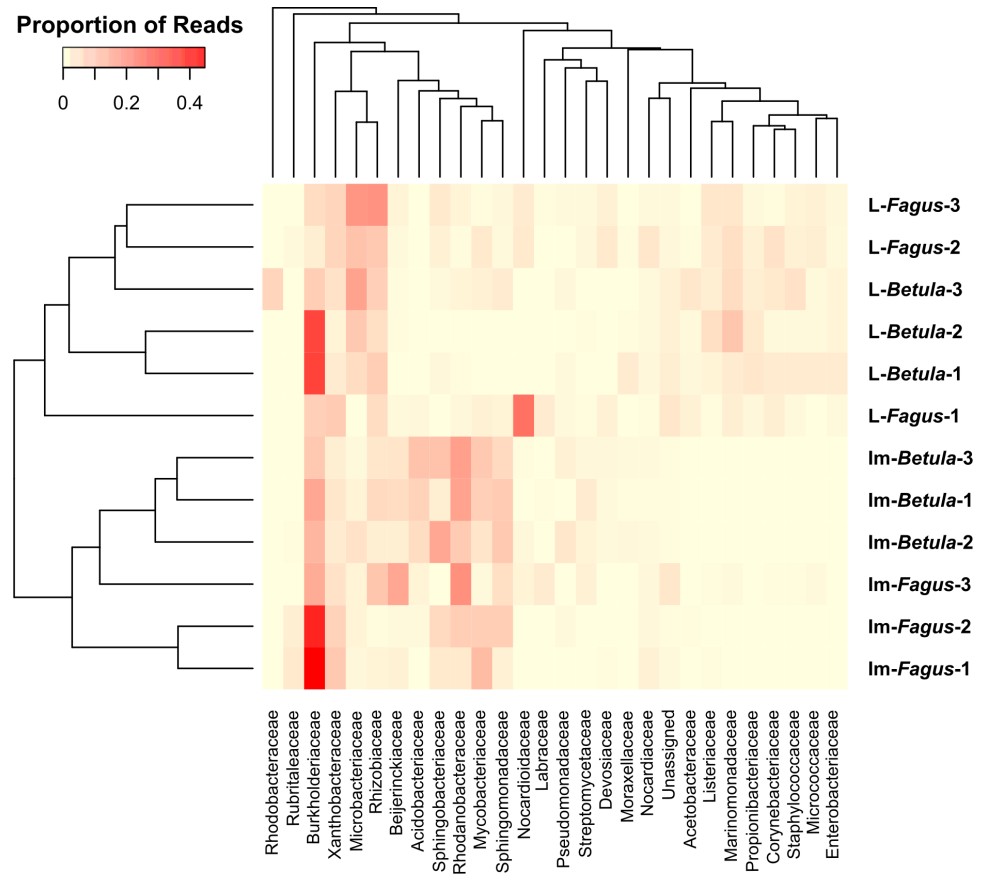

**Figure 3** **The heatmap showing bacterial families distributed across *B. reticulatus* samples.** Only those families which relative abundance was not less than 3% of at least one sample were considered. Cell values were calculated proportionately across rows and dendrograms were estimated with Bray-Curtis dissimilarity index.

**Table 2** **The level of inhibition, expressed in % inhibition of the marker enzyme, by protease inhibitors contained in *Fomes fometarius* fruiting bodies preparations.**

| pH | Marker enzyme | *Fagus* sp. | | *Betula* sp. | |
|---|---|---|---|---|---|
| | | Healthy fungus | Inhabited fungus | Healthy fungus | Inhabited fungus |
| 5.0 | pepsin | $6.20 \pm 0.03$ | 0.3 | $3.40 \pm 0.02$ | $0.10 \pm 0.01$ |
| 7.0 | papain | $0.20 \pm 0.01$ | 0 | $9.80 \pm 0.03$ | $5.70 \pm 0.03$ |
| | trypsin | 0 | 0 | $2.50 \pm 0.02$ | $0.20 \pm 0.01$ |
| 9.0 | trypsin | $0.10 \pm 0.01$ | 0 | $0.10 \pm 0.01$ | 0 |

## DISCUSSION

In the present study, we investigated the microbiome profiles of larvae and adults of fungivorous beetle *B. reticulatus*. Obtained patterns of the abundance at both phylum and class level remained in congruence with previous studies. Proteobacteria, Actinobacteria and Bacteroidetes phyla, as well as Alpha- and Gammaproteobacteria classes have been

**Table 3** Retardation factor (Rf) of secondary metabolites spots contained in methanolic and aqueous preparations of fruiting bodies of *Fomes fometarius*. TLC plates were developed using the ethanol-water system and were visualized under a UV lamp at wavelengths of 254 and 365 nm.

| Extract | UV wavelength (nm) | *Fagus* sp. | | *Betula* sp. | | |
|---|---|---|---|---|---|---|
| | | Healthy fungus | Inhabited fungus | Healthy fungus | Inhabited fungus | |
| Methanol | 365 | 0.8235 + | 0.8235 + | 0.8235 + | 0.8235 + | |
| | 254 | 0.7058 ++ | 0.7058 + | 0.7058 + | 0.7058 + | b |
| | 254 | 0.1176 +++ | – | – | – | |
| Water | 365 | 0.9092 + | – | – | – | |
| | 254 | 0.7882 ++ | – | – | – | a |
| | 254 | 0.7058 + | 0.7058 + | 0.7058 + | 0.7058 + | b |

**Notes.**

+, ++, +++, intensity of UV spots.
[a] catechins and their derivatives.
[b] sesquiterpenes lactones.

listed as the most abundant bacterial groups in various insect species (e.g., *Colman, Toolson & Takacs-Vesbach, 2012*; *Jones, Sanchez & Fierer, 2013*; *Yun et al., 2014*; *Kim et al., 2017*), and also in those cultivating fungi (*Aylward et al., 2014*). Moreover, identified patterns of the abundance remained in congruence with the results of our previous study focused on microbial communities associated with *Hoplothrips carpathicus* (Thysanoptera), which also inhabits fruiting bodies of *F. fomentarius* (*Kaczmarczyk et al., 2018*).

Interestingly, at the genus level *Burkholderia-Cabalerrinia-Paraburkholderia* (Burkholderiaceae) was one of the most dominant genera in all tested samples. In the study of bacterial communities associated with fungivorous *H. carpathicus* this genus was also noted (*Kaczmarczyk et al., 2018*), but it was not as abundant as in *B. reticulatus*. *Burkholderia* was also identified in bacterial communities associated with other insects e.g., in the larvae of the wood-feeding beetle *Prionoplus reticularis* (*Reid et al., 2011*), in longicorn beetle *Prionus insularis* (*Park et al., 2007*) or in members of Heteroptera (*Kikuchi, Hosokawa & Fukatsu, 2010*). This genus is linked with several functions: nitrogen fixation (*Estrada-De et al., 2001*), defence mechanisms (*Santos et al., 2004*), aromatic compound degradation (*Laurie & Lloyd-Jones, 1999*; *Bugg et al., 2011*), and detoxification of tree defence compounds (*Smith et al., 2007*; *Adams et al., 2013*). Furthermore, a symbiotic relationship between *Burkholderia* and white rot fungus *Phanerochaete chrysosporium* was described by *Seigle-Murandi et al. (1996)*. This fungus species, similar to *F. fomentarius*, degrades lignocellulosic materials. Nevertheless, symbiotic relationships between *F. fomentarius* and microorganisms has not been investigated yet. Therefore, one may not exclude that *Burkholderia* identified in bacterial communities of *B. reticulatus* is connected also with the tinder fungus via a symbiotic relationship. However, its presence in microbiome profiles of different developmental stages of black tinder fungus beetle is thought to be related to the potential of *Burkholderia* representatives to degradation of aromatic compounds (e.g., lignin) present in *B. reticulatus* food source.

Besides *Burkholderia*, we found in tested bacterial communities also other genera, which have been considered as degraders of aromatic compounds. *Pseudomonas*

(Pseudomonadaceae) and *Serratia* (Enterobacteriaceae) identified previously in microbiome of the mountain pine beetle *Dendroctonus ponderosae*, have been described as terpene degraders (*Adams et al., 2013*). In turn, *Sphingomonas*, *Sphingobium*, *Novosphingobium* and *Sphingopyxis* (Sphingomonadaceae) have been recognised being involved in degradation of various recalcitrant aromatic compounds and polysaccharides (*Aylward et al., 2013*). Moreover, *Sphingomonas* has been identified in microbiome of wood-boring beetle, *Anoplophora glabripennis* as genus involved in the degradation of lignocellulose, hemicellulose, and other aromatic hydrocarbons (*Geib et al., 2009a*; *Geib et al., 2009b*; *Scully et al., 2013*). *Bosea* (Beijerinckiaceae; also found in present study) was in turn identified being associated with cuticles of plant-ant genera *Allomerus* and *Tetraponera* (*Seipke et al., 2013*) and described as hydrocarbon degrader (*Yang et al., 2016*). Comprehensive analyses of the functional microbiome of arthropods (e.g., honeybees, fruit flies, cockroaches, termites, ants and beetles) show that *Burkholderia*, *Sphingomonas* and *Bosea* are together involved in the same processes e.g., aerobic metabolism, reacting with cytochrome *c* or bypassing cytochrome *c* (*Esposti & Romero, 2017*).

The bacterial community associated with black tinder fungus beetle likely plays a role in promotion of efficient digestion for extraction of maximum energy from ingested substrates. Nevertheless, specific conditions in microenvironment of *F. fomentarius* fruiting body may cause strong selective pressure against microorganisms that are not able to survive exposure to defensive compounds produced by the tinder fungus. Recent study shed light on antimicrobial activities of *F. fomentarius*. *Kolundžić et al. (2016)* found that the tinder fungus extracts of different polarity exhibit significant antimicrobial activity against nine bacterial strains (including *Staphylococcus aureus*, *Staphylococcus epidemidis*, *Bacillus subtilis* or *Klebsiella pneumonia*). In fact, the relative abundance of those bacteria in microbiome of *B. reticulatus* were low and in most samples tested did not exceed 1%. The observed antimicrobial activity of *F. fomentarius* may be linked e.g., with polyphenols and $\beta$-glucans which abundances are relatively high in its fruiting bodies (*Seniuk et al., 2011*; *Zhao et al., 2013*; *Alves et al., 2013*; *Zhu et al., 2015*) or with sesquiterpens which have been described as active compounds (identified enzyme inhibitors with antifungal, antibacterial and cytotoxic activities) (*Abraham, 2001*; *Keller, 2018*). These compounds may be considered as toxins for bacteria associated with fungivorous species inhabiting *F. fomentarius* fruiting bodies. Thus, associated bacterial communities need to overcome the presence of such substances through resistance or tolerance mechanisms.

Preliminary analyses performed in PICRUSt (*Langille et al., 2013*; Fig. S1) showed that in all tested bacterial communities genes involved in membrane transport and in metabolism of terpenoids and polyketides, as well as in xenobiotic biodegradation and metabolism should be elevated. Interestingly, terpenoids are main secondary metabolites of *F. fomentarius* (around 75%) (*Grienke et al., 2014*) what means that microorganisms associated with *F. fomentarius* feeding species remain exposed to these compounds. Moreover, the similar pattern of relative abundances were identified for predicted genes in the case of bacterial communities associated with thrips *H. carpathicus* (*Kaczmarczyk et al., 2018*). This might mean that similarities in predicted patterns of relative abundances of genes are characteristic for bacterial communities associated with fungivorous species

inhabiting fruiting bodies of wood-decaying fungi. However, the *in silico* predicted functions need to be validated *in vitro* in future studies.

The additional aim of the present study was connected with the long-term field observations showing that there is a difference between the amount of healthy and colonized sporocarps growing on the trunks of both beech and birch (Table S1). In the case of beech, the *F. fomentarius* fruiting bodies inhabited by insects were collected less frequently (~27% on average, in the range from 19% to 33% of all collected fruiting bodies growing on selected trunks). In turn, the number of colonized sporocarps growing on birch was higher (~66% on average) and ranged from 46% to 85% of all collected fruiting bodies growing on these trees. To identify the potential factors related to the observed differences in degree of fruiting bodies colonization by insects, we identified trends in biochemical profiles of fruiting bodies growing on two different tree hosts.

We analyzed the level of protease inhibitors and secondary metabolites detected in *F. fomentarius* fruiting bodies, which are involved in protection against fungivorous insects (*Anke & Sterner, 2002*; *Sabotič, Ohm & Künzler, 2016*). Moreover, we investigated the bacterial communities of larvae and adults of *B. reticulatus* for potential differences which could be related to host-tree species.

The biochemical analysis showed that a higher level of protease inhibitors was observed in healthy fruiting bodies than in colonized ones. Unfortunately, no previous research has been done to investigate this phenomenon, so it is unclear why inhibitor levels are lower in colonized fruiting bodies. Changes in levels of acidic, neutral, and alkaline proteases in colonized sporocarps may resulted from the inherent properties of insect proteases, which are dominated by serine and cysteine proteases over aspartic proteases (*Terra & Ferreira, 1994*). In turn, a high level of acid inhibitors of aspartic proteases may be additionally associated with the protection of fruiting bodies against pathogenic and saprophytic microscopic fungi (mainly molds) (*Monod et al., 2002*), which have high levels of aspartic proteases in their proteolytic apparatus.

Moreover, performed analyses indicated the presence of unidentified secondary metabolites in samples of non-colonized fruiting bodies collected from beech. Probably, these substances are able to determine the susceptibility of fruiting bodies to be colonized by insects and, generally, to be infected. It is worth noting that even dozen-year-old specimens of *F. fomentarius*, growing on beech wood, are usually completely healthy, while fungi growing on the birch, are colonized by insects and has signs of the presence of pathogenic fungi (e.g., mold). Therefore, some obvious contributing factors to this phenomenon (e.g., size of fruiting body, sun exposure or age) seem to be insignificant for determining the degree of fruiting bodies colonization by insects. Some authors showed that *Fagus sylvatica* produces flavonoids and organic acids which can be classified as repellents against insects (*Harborne, 1997*; *Simmonds, 2003*; *Treutter, 2005*; *Podgórski & Podgórska, 2009*). The accumulation of some flavonoids in the fruiting bodies of *F. fomentarius* could be a barrier against the fungivorous beetles. At this stage, it cannot be excluded that flavonoids are among those unidentified secondary metabolites detected in present study. However, the identification of the secondary metabolites and their potential impact on the relationship among host tree, fungi and insects require in-depth studies.

The biochemical analyses presented here should be treated with caution since the fruiting bodies sampled for these analyses were collected earlier than those from which *B. reticulatus* specimens were collected. Both sets of samples were collected for different studies. Although the obtained patterns needs to be tested in more complex studies where beetles would be collected from sporocarps, which would then be used for further biochemical analyses, the results presented here could be considered as insight in potential trends. More advanced biochemical analyses (e.g., using liquid chromatography or mass spectrometry) may provide a more complete insight into the biochemical profile of fruiting bodies. However, despite the observed trends in biochemical profiles, tested microbiome samples primarily clustered by developmental stage of *B. reticulatus* and host tree did not appear to impact the taxonomic distribution of the communities, what was supported by statistical analyses.

Additionally, known endosymbionts have been identified in microbiome profiles of *B. reticulatus*. Among them *Wolbachia* should receive a special attention. It is a well-known endosymbiont, which is estimated to be present in more than 65% of all insect species (*Hilgenboecker et al., 2008*; *Lewis & Lizé, 2015*). *Wolbachia* is related to five commonly recognized manipulation schemes: feminization, parthenogenesis induction, early and late male killing, and cytoplasmic incompatibility (*Engelstädter & Hurst, 2009*). It appears that these phenomena do not occur in *B. reticulatus*, but more comprehensive studies should be performed to test this hypothesis.

## CONCLUSIONS

In conclusion, this paper presents the insight into bacterial communities associated with two developmental stages of *B. reticulatus* beetle with the use of 16S rRNA sequence data. The approach based on NGS technique allowed us to characterize of tested microbiome. Moreover, it is the first approach to identification of factors which can be related to differences in degree of fruiting bodies colonization by insects. Results of this study show biochemical differences in fruiting bodies collected from birch and beech. We compared these results with those obtained during analyses of bacterial communities associated with *B. reticulatus*. However, the host-tree appears to have no effect on the bacterial communities associated with tested developmental stages of *B. reticulatus*. Despite the observed trends in biochemical profiles of sporocarps collected from both tree species, tested samples primarily clustered by developmental stage of *B. reticulatus*. Moreover, endosymbiotic Alphaproteobacteria *Wolbachia* was identified for the first time in *B. reticulatus*.

### Funding
The authors received no funding for this work.

### Competing Interests
Sylwia Zielińska is an employee of the Phage Consultants (Gdańsk, Poland).

## Author Contributions

- Agnieszka Kaczmarczyk-Ziemba conceived and designed the experiments, performed the experiments, analyzed the data, contributed reagents/materials/analysis tools, prepared figures and/or tables, authored or reviewed drafts of the paper, approved the final draft.
- Grzegorz K. Wagner conceived and designed the experiments, performed the experiments, analyzed the data, prepared figures and/or tables, authored or reviewed drafts of the paper, approved the final draft.
- Krzysztof Grzywnowicz conceived and designed the experiments, contributed reagents/materials/analysis tools, authored or reviewed drafts of the paper.
- Marek Kucharczyk contributed reagents/materials/analysis tools, authored or reviewed drafts of the paper.
- Sylwia Zielińska analyzed the data, prepared figures and/or tables, authored or reviewed drafts of the paper.

## Field Study Permissions

The following information was supplied relating to field study approvals (i.e., approving body and any reference numbers):

Field studies were approved by the Ministry of the Environment in Poland (approval numbers: DLP-III-4102-21/1728/15/MD for field study in Poleski National Park and DPL-LLL-4102-609/1699/14/MD for field study in Roztocze National Park).

## DNA Deposition

The following information was supplied regarding the deposition of DNA sequences:

The NGS data are available at ENA (PRJEB23388).

## Data Availability

Kaczmarczyk-Ziemba, Agnieszka; Wagner, Grzegorz K.; Grzywnowicz, Krzysztof; Kucharczyk, Marek; Zielińska, Sylwia (2019): Detailed taxonomic analyses at different ranks for DNA. figshare. Figure. https://doi.org/10.6084/m9.figshare.7928144.v1.

## Supplemental Information

Supplemental information for this article can be found online at http://dx.doi.org/10.7717/peerj.6852#supplemental-information.

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
