# Peer review of "The microbiome profiling of fungivorous black tinder fungus beetle Bolitophagus reticulatus reveals the insight into bacterial communities associated with larvae and adults"

_PeerJ, doi:10.7717/peerj.6852_

## Round 0.1 · original submission · Major Revisions

Dear Dr. Kaczmarczyk-Ziemba and colleagues:

Thanks for submitting your manuscript to PeerJ. I have now received three independent reviews of your work, and as you will see, the reviewers raised substantial concerns about the research. Nonetheless, there is optimism in two of the reviews, so I encourage you to revise your work based on the reviewer’s comments and resubmit your manuscript. I am sure that addressing these concerns will greatly improve your manuscript.

Please ensure that an English expert has thoroughly evaluated your revised manuscript.

In your revision, please consider all of the suggestions by the reviewers regarding your experimental design and data analyses. Please also provide justification for why certain methodological approaches were selected over others. Please consider some of the alternative approaches suggested by the reviewers.

Accordingly, I am recommending that you revise your manuscript, taking into account all of the issues raised by the reviewers. I look forward to seeing your revision, and thanks again for submitting your work to PeerJ.

Good luck with your revision,

-joe

Reviewer 1 ·

Basic reporting

The writing is good overall. Please check the formatting of the references, and confirm that all citations appear in the references section.

Experimental design

The design is good overall, though it seems several replicates for surveying the beetle microbiome were pooled. It perhaps would have been nice to sequence these separately to have true replicates, but overall the samples all looked quite similar.

Validity of the findings

The phylum-level classifications presented throughout the paper and in most of the figures are difficult to interpret given the vast diversity of prokaryotes within each phylum. It would be much more useful to see what bacterial families and genera were most abundant in each of the samples.
Figure 4 in particular would be much more powerful if it included the most abundant genera, rather than phyla. Currently the clustering is uninformative since all of the samples look relatively similar.

Line 36: “for long-term field” -> “for the long-term field”

An important role that bacteria could be playing in these tree associated systems is the degradation of plant defense compounds such as terpenes, which can be toxic to fungi and beetles. Sphingomonads have previously been shown to have versatile biodegradative capacity and could be involved in this process (Aylward et al., Applied and Environmental Microbiology, 2013) and microbial communities associated with pine beetles have also been hypothesized to play this role (Adams et al, Applied and Environmental Microbiology, 2013). I would encourage the authors to emphasize this potential role a bit more, since it also fits with the presence of Burkholderia (as the authors point out in the paragraph starting on line 409).

Line 444: “The bacterial community associated with black tinder fungus beetle likely plays a role in promotion of efficient digestion for extraction of maximum energy from ingested substrates”. Is this really true? Is it possible that the bacterial community is mostly associated with the fungus, and we see it in the beetles simply because of their diet?

I do not think the PiCRUSt analysis adds anything to this study, and it could potentially be omitted to streamline the manuscript. The authors find no significant differences between pathways based on this analysis, and it is hard to interpret findings based on the potential relative abundances of broad functional categories like “membrane transport”. If the authors really wish to keep it, I would recommend just briefly mentioning that it was done and did not show any meaningful differences between samples.

It is unclear to me what the importance of protease inhibitors is in this system. The authors should consider describing the importance of this more in-depth in the introduction if they feel it is an important part of this study.

The discussion is quite long and speculative. I would consider limiting or omitting the PiCRUSt analysis and streamlining the discussion of protease inhibitors. By focusing on the dominant bacterial genera present I think the most important aspects will be conveyed.

Previous work has shown that bacteria of the genera Pseudomonas, Enterobacter, and Rahnella are most abundant in beetle and fungus-associated niches. Is this true here? It’s possible to see this from the supplementary sunburst diagrams, but it would be nice if this was more clearly mentioned in the Discussion.

Additional comments

Overall I think this paper would be suitable for publication if the authors address the comments above and streamline the paper so it is a bit more brief and directed.

Reviewer 2 ·

Basic reporting

The research paper submitted to PeerJ and entitled "Predictive functional profiling and surveying the microbiome of fungivorous black tinder beetle Bolitophagus reticulatus" by Kaczmarczyk-Ziemba et al deals with the metataxonomic analysis of the microbiome associated with the beetle Bolitophagus reticulatus and its predicitive functional profiling.

It almost solely relies on the predictive functional analyses using PiCrust and based on 16S phylogeny. This is too speculative in general and thus I would actually suggest to the authors to formally access the functional potential of microbial communities to perform a shotgun metagenomic analysis. Furthermore, the diversity analyses s quite shallow with only a venn diagram and barplots. This could well be improved.
I would also like to point out that the greengene database that was used to infer taxonomy has not been updated since 2013 so I would suggest the authors to use the Silva one for example.

So while the research question is of interest, the tools used are not completely appropriate.

And, this is why I tend to agree with the last sentence of the conclusion. This study in this form to my point of view is a pilot study and needs to be performed at a wider scale prior to publication.

Experimental design

As indicated above, the choice of approach to infer functional potential is not ideal as well as the database used to assess taxonomical affiliations.

The metagenomic DNA extraction was performed on the whole body of the insects. Could the authors maybe explain why they haven't extracted the gut microbiome, ie dissected the insects and larvaes? That would have been more specific and proper to answer their research question

The authors have performed a 'preliminary biochemical analyses' of sporocarps.
First, I'm unsure what this means (could it be improved? is it incomplete?) and second, why have they not done these analyses on the fruiting bodies collected for this 16S study? To my opinion this approach makes no sens as long as they have not shown that the biochemical analyses of the fruiting bodies studied is constant through time and space...And, why did they specifically choose sporocarps from fallen trees for the biochemical analyses and not for the 16S one?

Validity of the findings

no comment

Additional comments

Generally, I also find that having a 10p discussion for only 4p of result is too long and dilutes the scientific message that the authors are trying to make. I would thus really suggest the authors to shorten the discussion.
I would also recommend to rewrite the introduction significantly as it does not seem linked to the research questions. And it has some vague statements (eg L95-102).

Minor comments:
Often the authors refer to 'previous studIES' and only cite one (eg L70-73 or L91). Also the referencing in the text is not constant (eg L87 "Jonsell, Schroeder & Larson, 2003" should be "Jonsell et al, 2003"). All this should be fixed
Finally, while English is not my mother tongue, I strongly suggest the authors to ask for an English speaker to check their manuscript for minor, but existing, grammatical mistakes.
Scale bars on Figure 1 would be more than appreciated.
Abbreviations should be defined the first time they appear (eg NGS L105)à

·

Basic reporting

Kaczmarczyk-Ziemba and colleagues explore the microbiota of the fungivorous black tinder beetle, Bolitophagus reticulatus. They use 16S rRNA gene amplicon sequencing of 12 samples--3 from each developmental stage (larvae, adult) from each tree type (beech, birch)—to compare the microbiomes. Exploring the microbiota associated with fungivorous insects is interesting and highly warranted, as such studies will help lead towards a broader understanding of how insect microbiota is associated with different diets.

General Comments
• The grammar/syntax/flow could be improved (some examples included in specific comments below).
• PICRUSt is not very reliable for 16S amplicon sequencing. 16S only resolves taxonomic placement to the family/genus level, while gene context varies hugely within these genera and families of fungi. Yes, this is commonly published on in the literature, but that doesn’t make it reliable. In my opinion, this work doesn’t add anything to this study regardless, so I would just cut it all out.
• The breakdown of the bacterial composition of their samples is interesting, although there weren’t many differences. The issue is that the number of samples is rather small for a 16S amplicon study. Further, it is limited by the 16S. I do think the comparison to the work in Aylward et al. 2014 is interesting and a strength of this study; meaning it is interesting to compare beetles that feed on fungus that is cultivated or symbiotic, with beetles that are feeding on ‘free-living’ (with regards to the beetles) fungus.
• Rf values are stated but no compounds are talked about? They said in their methods the Rf values would be compared to Rf values in a database to distinguish compounds. Were they not able to distinguish any compounds from the Rf values?
• The statement that they “nearly fully characterize its microbiome” is a completely over the top. 12 samples with 16S amplicon sequencing is just pilot findings in my view.
• GreenGenes uses incomplete 16S sequences, making it error prone. Consider taxonomic assignment using the Silva database instead.


Specific Comments
• Lines 67-72: refers to “studies” 4 times
• 73: “gained the raising interest” should be reworded
• 74/76: same paper cited twice in the same sentence.
• 79: One of the most widely distributed beetles?
• 83: It’s not clear that tinder fungus is the same as Fomes fomentarius. I think this should be spelled out.
• 106: varies  vary
• 258: ranking ranging
• 259: what  which
• 278/285/293/392/394: Phyla and class aren’t italicized?
• 337-338: awkward wording
• 355: “these compound affect not only direct insects”  Not only do these compounds directly affect insects
• 421/424: Nevertheless used to start both sentences
• Discussion of Burkholderia (and other genera) was interesting and linked predicted function to potential function in vivo
• Protease inhibitors/microbiome sound like two separate projects and aren’t linked together well
- 586: exceed excess?
- Line 48: “extremely important” seems overstated, and “important” should generally be avoided in scientific writing as the word can come across presumptuous. Try “critical” or “consequential.”
- Line 50-56 can be shortened to: “Saproxylic beetles can occupy several ecological niches:…”
- Line 57-58 can be reworded to: “Mycetophilic beetles associate with wood-decaying fungi and use the fruiting body for nourishment and development.”
- Line 73: considering removing “the raising”
• Avoid starting a sentence or a clause with the word “it” because “it” is ambiguous in those contexts. Examples: Line 90, Line 332, 372, 507, 509, 517.
- Line 136: “the inhabiting” can be reworded to “inhabited”
- Line 308: “allowed to” can be reworded to “allowed us to”
- Line 480-483: phrasing makes comprehension difficult.
- Figure 5: y-axis could use a label.
- Line 167-170 could use brief clarification. How were the samples merged? Were the number of reads of each sample normalized before merging?
- Lines 79-88 are unclear. It states that B. reticulatus is widely distributed but that it was thought to have limited dispersal until another paper found that the dispersal ability may have been underestimated. It’s not clear if they are trying to link its fungal association with dispersal?
- Lines 96-102: They switch from talking about SM of tree being accumulated in fungus to SM produced by fungus. If the “however” in 95 was switched to “additionally” that suggests to me they are interested in investigating both tree SM and fungus SM.
- Unclear why they investigated protease inhibitors other than “little is known”
- Mixes adult stage and imago stage. Should either stick with one or make it clear they are the same developmental stage.
- Figure 3 needs to be labelled better since it was confusing what the difference between the two graphs was. I’m not sure the “zoom in” is necessary and this figure can probably just be the one stacked bar graph.

Experimental design

see above

Validity of the findings

see above

Additional comments

see above

---

## Round 0.2 · Minor Revisions

Dear Dr. Kaczmarczyk-Ziemba and colleagues:

Thanks for revising your manuscript based on the concerns raised by the reviewers. While your article is now scientifically acceptable there are still language issues which you should try to address before it enters production, so I am giving you a chance to submit a revision before final acceptance.

I look forward to seeing this work in print, and I anticipate it being an important resource for the communities studying beetle microbiomes. So please address these grammatical issues as soon as possible. Thanks again for choosing PeerJ to publish such important work.

Best,

-joe

Reviewer 1 ·

Basic reporting

Improved

Experimental design

Improved

Validity of the findings

Valid and interesting

Additional comments

I think the authors have done a good job addressing the points raised in the initial round of review. The new manuscript is much easier to read, much more streamlined, and much more effective and making key points. The figures are also easier to follow. I am not the best person to evaluate the results of the biochemical analyses, but it seems as though the authors have been very forthright in their Discussion about possible caveats here, so these seem like valid data to include.
I am recommending "minor revisions" mainly so there is a chance to change the grammatical issues I raise below. I have no further recommendations for changes to the scientific content.

Line 25: “One of the group among them are” -> “among this group are”
Line 30: “the detailed studies focused on microbiome associated” -> “detailed studies focusing on the microbiome associated with”
Line 69: “This beetle belongs to tribe Bolitophagini which represent feeding strategy of dwellers”. I’m not entirely sure what this sentence means, or how the term “dwellers” should be interpreted, but it should be re-worded for clarity. Perhaps, “This beetle belongs to tribe Bolitophagini, which represents the feeding strategy of dwellers, which refers to xxx”
Line 94: “Know” -> “Known”
I would avoid the term “preliminary” or “pilot” when referring the analyses here. These terms make it sound like the work isn’t ready to be published. If the authors are confident in the results, they can be presented as legitimate analyses along with all the rest. If there are any important problems with these analyses that the authors are aware of, they should just state those outright in the Discussion as potential complicating factors (I think the authors have done a good job of this in the paragraph starting on line 429).
Line 251: “approximately deep” -> “sufficient for microbiome characterization”
Line 289: “allow to identify” => “revealed”
Line 318: “listed as” -> “listed as the”
Line 320: “also those” -> “and also in those”
Line 429: “We are aware that these biochemical analyzes should be treated as preliminary. Fruiting bodies for biochemical analyses were collected earlier than those from which B. reticulatus specimens were collected” -> “The biochemical analyses presented here should be treated with caution since the fruiting bodies sampled for these analyses were collected earlier…”
Line 440: “Additionally, the known endosymbionts has” -> “Additionally, known endosymbionts have”

·

Basic reporting

see below

Experimental design

see below

Validity of the findings

see below

Additional comments

The authors addressed my previous comments.

---

## Round 0.3 · accepted · Accept

Dear Dr. Kaczmarczyk-Ziemba and colleagues:

Thanks for revising your manuscript based on the minor concerns raised by the reviewers. I now believe that your manuscript is suitable for publication. Congratulations! I look forward to seeing this work in print, and I anticipate it being an important resource for the groups studying the microbial diversity of beetles. Thanks again for choosing PeerJ to publish such important work.

Best,

-joe

#